# Underwater Target Localization and Synchronization for a Distributed SIMO Sonar with an Isogradient SSP and Uncertainties in Receiver Locations

**DOI:** 10.3390/s19091976

**Published:** 2019-04-27

**Authors:** Chaofeng He, Yiyin Wang, Wenbin Yu, Lei Song

**Affiliations:** The Department of Automation, Shanghai Jiao Tong University, Collaborative Innovation Center for Advanced Ship and Deep-Sea Exploration (CISSE), Key Laboratory of System Control and Information Processing, Ministry of Education of China, Shanghai 200240, China; cfhe@alumni.sjtu.edu.cn (C.H.); yuwenbin@sjtu.edu.cn (W.Y.); songlei_24@sjtu.edu.cn (L.S.)

**Keywords:** localization, clock synchronization, sound speed profile, uncertainties of locations, distributed sonar

## Abstract

A distributed single-input multiple-output (SIMO) sonar system is composed of a sound source and multiple underwater receivers. It provides an important framework for underwater target localization. However, underwater hostile environments bring more challenges for underwater target localization than terrestrial target localization, such as the difficulties of synchronizing all the underwater receiver clocks, the varying underwater sound speed and the uncertainties of the locations of the underwater receivers. In this paper, we take the sound speed variation, the time synchronization and the uncertainties of the receiver locations into account, and propose the underwater target localization and synchronization (UTLS) algorithm for the distributed SIMO sonar system. In the distributed SIMO sonar system, the receivers are organized in a star topology, where the information fusion is carried out in the central receiver (CR). All the receivers are not synchronized and their positions are known with uncertainties. Moreover, the underwater sound speed is approximately modeled by a depth-dependent sound speed profile (SSP). We evaluate our proposed UTLS algorithm by comparing it with several benchmark algorithms via numerical simulations. The simulation results reveal the superiority of our proposed UTLS algorithm.

## 1. Introduction

As a fundamental function for sonar systems, target localization based on multistatic sonar has gained significant attention. Multistatic sonar can provide superior localization accuracy and detection performance as well as improved robustness and flexibility over its monostatic counterpart [1]. There are several typical multistatic systems enabling target localization, such as multi-input multiple-output (MIMO) sonar/radar [2,3], single-input multiple-output (SIMO) sonar/radar [4,5], wireless sensor networks [6] and multistatic radar [7,8]. Multiple trials conducted by the Centre for Maritime Research and Experimentation [9] have shown that multistatic sonar systems can provide better detection, localization and tracking performances than monostatic sonars. In this paper, we focus on the underwater target localization based on the SIMO sonar system which consists of a sound source and multiple receivers organized in a star topology.

Various localization algorithms have been proposed for wireless networks [10,11,12], and terrestrial SIMO or MIMO radar systems [13,14]. However, they cannot be applied to the distributed sonar system directly due to the special challenges posed by underwater environments. Firstly, the underwater sound ray (USR) bends according to the varying underwater sound speed. The curve of the USR gives rise to ranging errors thereby localization errors. Secondly, The mobility of the underwater receivers results in uncertainties of the receiver locations. Therefore, the real locations of the receivers may not be the same as what we have known. Thirdly, time synchronization has to be carried out due to the absence of GPS and clock differences of underwater receivers. Equipping the underwater receivers with accurate atomic clocks [15] is a way to keep all the underwater receiver clocks in synchronization. However, the atomic clock is quite expensive [16]. Various time synchronization algorithms have been proposed for underwater devices [17,18]. However, most of them require a number of signal transmissions which are energy intensive. In addition, it is difficult to replace batteries for underwater receivers due to the hostile underwater environments. Therefore, it is necessary to perform target localization considering the variation of underwater sound speed, the uncertainties of the receiver locations and the time synchronization.

The sonar researchers have proposed a lot of localization algorithms for various sonar systems [19,20]. The authors of [21] perform a fundamental investigation on the performance of elliptic localization. In [22], the authors utilize the time difference of arrival- angle of arrival (TDOA-AOA) measurements obtained by two stations to localize a point source. A Bayesian localization method for multistatic active sonar is proposed in [23]. The authors of [24] take the mobility of the target into consideration and propose a moving target localization algorithm for multistatic sonar. The multi-target localization problem is considered in [25], the authors propose a multi-target positioning algorithm via bistatic range space projection. Efficient closed-form estimators are proposed for multistatic sonar localization in [26].

As another type of underwater target localization, underwater source localization is also a hotspot [27,28,29,30,31,32,33,34,35,36,37,38,39,40,41,42,43]. The authors of [27] implement source localization from the sparse signal reconstruction perspective. A lecture note of source localization from range-difference measurements is shown in [28]. In [29], the TDOA and gain ratios of arrivals are combined to perform passive source localization. The authors of [30] propose a nonparametric iterative adaptive approach for source localization and sensing.

Some works take the location uncertainties of the anchors into consideration [31,32,33,34,35,36,37,38] in source localization. The authors of [31,32,33] analyse the effect of anchor location errors on source localization. In [34,35], the calibration emitters are utilized to alleviate the effect of anchor location errors on source localization. The authors of [36] consider the problem of locating multiple disjoint sources using TDOAs and frequency differences of arrival (FDMA) in the presence of anchor position and velocity errors, and propose an asymptotically efficient estimator. The synchronization clock bias and sensor position errors are considered in the TDOA source localization in [37]. In [38] a robust localization algorithm using geographic information in bistatic sonar is proposed for source localization. Some researchers attempt to find a close-form solution for underwater source localization [39,40,41,42,43]. Some closed-form solutions for source localization are obtained by utilizing the TDOA measurements [42] or the combination of TDOA and FDMA measurements [40]. The problem of source localization is extended to multiple disjoint sources localization in [41], and an approximately efficient closed-form solution is proposed. In [43], the authors proposed a closed-form solution for joint source and sensor localization. It locates multiple disjoint sources and refines erroneous sensor positions simultaneously.

All the above underwater target localization algorithms assume that the underwater sound speed is constant during the process of localization. The constant sound speed results in the straight-line trajectory of the USR. However, the assumption of the constant underwater sound speed is validated only in some shallow sea environments. Generally, the underwater sound speed varies with depth, temperature, salinity, etc. Especially, it can be modeled by an isogradient depth-dependent sound speed profile (SSP) approximately in deep sea environments [44] as the salinity and temperature almost remain unchanged. Therefore the depth is the only factor that affects the SSP in deep sea. In deep sea environments, the variation of the sound speed makes the trajectory of the USR a curve. Therefore, the constant sound speed assumption makes the performance degradation of underwater localization in deep sea environments.

The authors of [45,46] proposed algorithms to perform localization and tracking of a mobile target with an isogradient SSP. however, the time synchronization is ignored by them. A target localization algorithm is proposed for a distributed SIMO sonar with an isogradient SSP in our previous work [4]. A delay model is presented in an explicit form with respect to (w.r.t.) the locations of receivers and target. The clock parameters of the receivers and the location of the target are estimated with the assumption that the locations of the receivers are accurate. Actually, in order to monitor efficiently, the receivers have to be anchored at some special depths with anchor chains. In this case, the receivers drift with ocean currents. Therefore, the locations of the receivers vary with ocean currents, which means that there are uncertainties in the receiver locations.

In this work, with the consideration of the USR curve, the time synchronization and the uncertainties of the receivers, we propose the underwater target localization and synchronization (UTLS) algorithm for the distributed SIMO sonar system. The receivers consist of one central receiver (CR) and several normal receivers (NRs) organized in a star topology. Assuming that the receivers are anchored at some special depths with anchor chains, all the receivers are not synchronized, and the underwater sound speed is modeled by an isogradient SSP.

We assume that all the receivers are equipped with acoustic modems [47] to communicate with each other. The CR starts the synchronization and target localization process by broadcasting an initial signal to the NRs after the reception of the direct and reflected signals. In the UTLS algorithm, we first employ the expectation maximization (EM) algorithm [48] to estimate the clock skews of the receivers, and apply the weighted least squares (WLS) to estimate the clock offsets sequentially with the clock skews estimated via the EM algorithm. During the estimation of the clock skews, the receiver locations are estimated as latent variables with the consideration of the uncertainties of the receiver locations. We apply the nonlinear weighted least squares (NWLS) to perform target localization with the estimates of the clock parameters and the receiver locations. Simulation results show the superiority of the UTLS algorithm compared to the tailored benchmark algorithms, such as the tailored NWLS algorithms, the algorithms which combine the tailored WLS and NWLS algorithms.

The rest of the paper is organized as follows. In Section 2, we describe the system models which include the sound speed and time delay models, the clock model, the uncertainty model of the receiver locations and the measurement models. In Section 3, the UTLS algorithm is introduced. It includes the EM algorithm, the WLS algorithm and the NWLS algorithm to achieve time synchronization and target localization. We evaluate the performance of the UTLS algorithm by comparing it with the tailored benchmark algorithms in Section 4 through several simulations. Finally, we conclude this paper in Section 5.

## 2. System Models

The scenario considered in this research consists of one target whose coordinates are $x={\left[x,y,z\right]}^{T}$, one sound source and M+1 distributed receivers $\left\{r_{i},i=1,2,\cdots ,M,h\right\}$ as shown in Figure 1. The receivers organized in the fashion of a star topology. The receiver $r_{h}$ is referred as CR, while the receivers $r_{i},i=1,2,\cdots ,M$ are referred as NRs. The coordinates of the receivers are indicated as $\left\{x_{i},i=1,2,\cdots ,M,h\right\}$. We do not know the accurate locations of the receivers as they move with ocean currents. The signals depart from the sound source directly to the receivers $r_{i},i=1,2,\cdots ,M,h$ are referred as direct signals $s^{d}=\left\{s_{1}^{d},\cdots ,s_{M}^{d},s_{h}^{d}\right\}$; while the signals which depart from the sound source, reflected by the target and received by the receiver $r_{i},i=1,2,\cdots ,M,h$ are referred as the reflected signals $s=\{s_{1},\cdots ,s_{M},s_{h}\}$. There are a large amount of research focus on the detection of the direct signals and the reflected signals [49,50,51]. We assume that the detection problem can be solved by the existing approaches, and focus on the target localization and time synchronization algorithm.

In order to estimate the target location x, there are three aspects have to be considered. The first one is the varying underwater sound speed which results in the curve of USR. The second one is time synchronization. The third one is the uncertainties of the locations of the receivers. With the consideration of these three aspects, we employ four types of models: the sound speed and time delay models, the clock model, the uncertainty model of the receiver locations, and the time-of-arrival (TOA) measurement models.

### 2.1. The Sound Speed and Time Delay Models

The first aspect relates to the USR bend. As we know, the underwater sound speed is not constant but varies with temperature, salinity, pressure (depth), etc. The SSP can be approximated as a piece-wise linear function of the depth [52]. In our scenario, the assumption of isogradient SSP is made. Therefore, the SSP c(z) can be expressed as(1)$$\begin{array}{c} c(z)=az+b, \end{array}$$where *a* is a constant, and *b* is the sound speed at the surface. Both of them are assumed to be known as a priori. For a single ray such as the ray from the target to the receiver $r_{i}$ as shown in Figure 1, the time-of-flight (ToF) [4] $\tau_{i}$ can be expressed in an explicit form w.r.t. x and $x_{i}$(2)$$\begin{array}{cc} \tau_{i}≜f_{i}\left(x,x_{i}\right)= & \frac{1}{a}\ln\left\{\frac{{\left[ad_{i}+\sqrt{a^{2}d_{i}^{2}+4c\left(z_{i}\right)c\left(z\right)}\right]}^{2}}{4c\left(z_{i}\right)c\left(z\right)}\right\},\\ i= & 1,2,\cdots ,M,h, \end{array}$$where $d_{i}=\sqrt{{\left(x_{i}-x\right)}^{2}+{\left(y_{i}-y\right)}^{2}+{\left(z_{i}-z\right)}^{2}}$ denoting the distance between the receiver $r_{i}$ and the target. Similarly, the delays $\tau_{hi}$ between the CR and the NRs can be expressed as(3)$$\begin{array}{cc} \tau_{hi}≜f_{hi}\left(x_{h},x_{i}\right)= & \frac{1}{a}\ln\left\{\frac{{\left[ad_{hi}+\sqrt{a^{2}d_{hi}^{2}+4c\left(z_{i}\right)c\left(z_{h}\right)}\right]}^{2}}{4c\left(z_{i}\right)c\left(z_{h}\right)}\right\},\\ i= & 1,2,\cdots ,M, \end{array}$$where $d_{hi}$ is the distance between the CR and the *i*th NR. The delay $\tau_{s}$ between the sound source and the target can be written as(4)$$\begin{array}{cc} \tau_{s}≜f_{s}\left(x\right)= & \frac{1}{a}\ln\left\{\frac{{\left[ad_{s}+\sqrt{a^{2}d_{s}^{2}+4c\left(z_{s}\right)c\left(z\right)}\right]}^{2}}{4c\left(z_{s}\right)c\left(z\right)}\right\}, \end{array}$$where $d_{s}$ is the distance between the sound source and the target. The delays $\tau_{i}^{d}$ between the sound source and the receivers can be expressed as(5)$$\begin{array}{cc} \tau_{s}^{d}≜f_{is}\left(x_{i}\right)= & \frac{1}{a}\ln\left\{\frac{{\left[ad_{is}+\sqrt{a^{2}d_{is}^{2}+4c\left(z_{s}\right)c\left(z_{i}\right)}\right]}^{2}}{4c\left(z_{s}\right)c\left(z_{i}\right)}\right\}. \end{array}$$

### 2.2. The Clock Model

The second aspect is the time synchronization. In this work, we assumed that all the NRs were fully asynchronous. The clock model can be stated as(6)$$\begin{array}{cc} T_{i} & =\mu_{i}t+o_{i},\;i=1,2,\cdots ,M,h, \end{array}$$where $\mu_{i}$ and $o_{i}$ are the unknown clock skew and offset of the receiver $r_{i}$, respectively. The symbol $T_{i}$ stands for the timestamp from the clock of the receiver $r_{i}$ w.r.t. standard time *t*. For simplicity, we stack all the clock skews and clock offsets into the vectors $\mu ={\left[\mu_{1},\mu_{2},\cdots ,\mu_{M},\mu_{h}\right]}^{T}$ and $o={\left[o_{1},o_{2},\cdots ,o_{M},o_{h}\right]}^{T}$, respectively.

### 2.3. The Uncertainty Model of the Receiver Locations

The uncertainties of the receivers are caused by ocean currents and measurement noise. The ocean current pushes the receivers to move within some ranges. It makes the real locations of the receivers unknown. We obtain the receiver locations by measuring the ocean current. However, the measurements are corrupted by noise. Therefore we have to carry out time synchronization and target localization with the corrupted locations of the receivers.

#### 2.3.1. The Ocean Current Model

The double-gyre phenomena in large-scale ocean circulation is typical for the northern mid-latitude ocean basins [53]. It is quite dominant and persistent in oceans and consists of a sub-polar and a sub-tropical gyre. However, the double-gyre model is only suitable for large-scale ocean circulation. It cannot resolve local, small-scale flow disturbances properly. In addition to the double-gyre model, a small-scale gyre model is adapted in [53] to depict the local, small-scale background flow velocity (BFV) disturbances. In this paper, we adopt the ocean current model used in [53](7)$$\begin{array}{c} v\left(t\right)=v_{0}\left(t\right)+v_{1}\left(t\right), \end{array}$$where $v_{0}\left(t\right)$ and $v_{1}\left(t\right)$ are the large-scale and small-scale BFV model, respectively. We refer to [53] for more details of the ocean current model.

#### 2.3.2. The Positions of Receivers

Note that the BFV relates to the position. We denote the BFV at the position of the *i*th receiver as $v_{i}\left(t\right)$. In order to keep expressions concise, the time parameters of the BFV and the positions of the receivers are omitted when it dose not cause ambiguity. There are some assumptions given by

In order to monitor the ocean more efficiently, the receivers need deploying at some special depths. Therefore, we assume that all the receivers are anchored at the bottom of the sea with anchor chains. The length of the *i*-th receiver chain is $\zeta_{i}$.All the receivers are equipped with buoyancy balls to keep them suspending underwater and straightening the anchor chains.All the receivers are equipped with pressure sensors to determine their depths.All the receivers can measure the BFVs around their positions by using an acoustic Doppler current profiler.The initial position of the *i*-th receiver is the receiver’s position when the anchor chain is perpendicular. The initial positions are known as prior information.

With these assumptions, the feasible positions of the *i*th receiver compose a hemispherical surface as shown in Figure 2. Actually, the receivers drift with the BFV, their positions determined by the BFV. As the buoyancy balls always keep the receivers suspending underwater, the depths of the receivers are determined by the strength provided by the BFV. Their horizontal coordinates are determined by the directions of the BFV. The receiver $r_{i}$ first measures its depth, then utilizes the depth measurement to calculate the horizontal distance between its initial position and its current position. The horizontal distance given by(8)$$\begin{array}{c} r_{i}=∥\left[x_{i},y_{i}\right]-\left[x_{i}\left(0\right),y_{i}\left(0\right)\right]∥=\sqrt{\zeta_{i}^{2}-{\left(\zeta_{i}-z_{i}+z_{i}\left(0\right)\right)}^{2}}, \end{array}$$where ${\left[x_{i},y_{i},z_{i}\right]}^{T}$ and ${\left[x_{i}\left(0\right),y_{i}\left(0\right),z_{i}\left(0\right)\right]}^{T}$ are the positions of the receiver $r_{i}$ at time *t* and 0, respectively. The parameter $r_{i}$ is the horizontal distance between the positions of the receiver $r_{i}$ at time *t* and 0. With the horizontal distance, the feasible positions *i*th receiver are limited to a circle whose center is $\left[x_{i}\left(0\right),y_{i}\left(0\right),z_{i}\right]$ and radius is $r_{i}$. As shown in Figure 3, the receiver drifts with the ambient BFV, the direction of the BFV is the direction of the horizontal coordinates of the receiver. Therefore the horizontal coordinates can be given by$$\begin{array}{c} {\left[x_{i},y_{i}\right]}^{T}={\left[x_{i}\left(0\right),y_{i}\left(0\right)\right]}^{T}+\frac{r_{i}}{∥v_{i}∥}v_{i}. \end{array}$$where $v_{i}$ is the ambient BFV of the receiver $r_{i}$. In conclusion, by combining the horizontal coordinates with the depth coordinate, we have(9)$$\begin{array}{c} x_{i}=x_{i}\left(0\right)+{\left[\frac{r_{i}}{∥v_{i}∥}v_{i}^{T},z_{i}-z_{i}\left(0\right)\right]}^{T}. \end{array}$$

Actually, it is hard to obtain the real BFV and the real depth of the receiver. We only have the corresponding measurements which are corrupted by measurement noise. Taking the measurement noises into account we have(10)$$\begin{array}{c} {\overset{ˇ}{x}}_{i}=x_{i}+n_{x_{i}},i=1,2,\cdots ,M,h, \end{array}$$where $n_{x_{i}}={\left[n_{ix},n_{iy},n_{iz}\right]}^{T}$ is assumed as a zero-mean Gaussian noise with variance $\sigma_{x_{i}}^{2}=\sigma_{x}^{2}1_{3},i=1,2,\cdots ,M,h$, and the vector ${\overset{ˇ}{x}}_{i}$ is a measurement of $x_{i}$.

### 2.4. The TOA Measurement Models

Figure 4 reveals the signal time sequence of the target localization process. Firstly, a signal is transmitted by the sound source at time $t_{0}$, and arrives at the target at time $t_{0}+\tau_{s}$. We assume that the signal is reflected by the target as soon as it arrives at the target. Secondly, the direct signal $s_{i}^{d}$ received by receiver $r_{i}$ at time $t_{i}^{d}=t_{0}+\tau_{i}^{d}$. The receiver $r_{i}$ records the arrival time of the direct signal $s_{i}^{d}$ as $T_{is}^{d}$. The measurement model of the direct signals can be stated as(11)$$\begin{array}{c} T_{is}^{d}=\mu_{i}\left(t_{0}+\tau_{i}^{d}\right)+o_{i}+n_{i}^{d},i=1,2,\cdots ,M,h, \end{array}$$where $n_{i}^{d}∼N\left(0,\sigma_{i}^{2}\right),i=1,2,\cdots ,M,h$ are Gaussian measurement noises of the direct signals $s^{d}$.

Thirdly, the reflected signal $s_{i}$ received by the receiver $r_{i}$ at time $t_{i}=t_{0}+\tau_{s}+\tau_{i}$. The receiver records the arrival time of the reflected signal as $T_{is}$. The measurement model of the reflected signal can be represented as(12)$$\begin{array}{cc} T_{is} & =\mu_{i}\left(t_{0}+\tau_{s}+\tau_{i}\right)+o_{i}+n_{i},\\ i & =1,2,\cdots ,M,h, \end{array}$$where $n_{i}∼N\left(0,\sigma_{i}^{2}\right)$ is the Gaussian measurement noise of the reflected signal $s_{i}$ with $\sigma_{i}^{2}$ known as a priori. The measurement noises of each receivers are assumed to be independent with each other. Therefore, the elements of the set $\left\{n_{i},n_{i}^{d},i=1,2,\cdots ,M,h\right\}$ are independent with each other.

Fourthly, the CR broadcasts a signal at time $t_{hb}$ to start the information gathering process and records the broadcast time as $T_{hb}$. The signal arrives at the NR $r_{i}$ at time $t_{hb}+\tau_{hi}$, where $\tau_{hi}$ is the propagation delay between the CR and the NR $r_{i}$. The NR $r_{i}$ records the signal arrival time and denotes it as $T_{ih}$. The measurement models of the signals from the CR to the NRs are expressed as(13)$$\begin{array}{cc} T_{hb} & =\mu_{h}t_{hb}+o_{h}, \end{array}$$
(14)$$\begin{array}{cc} T_{ih} & =\mu_{i}\left(t_{hb}+\tau_{hi}\right)+o_{i}+n_{ih},i=1,2,\cdots ,M, \end{array}$$where $n_{ih}∼N\left(0,\sigma_{i}^{2}\right),i=1,2,\cdots ,M$ are independent Gaussian measurement noises.

Fifthly, the NR $r_{i}$ replies a signal to the CR at time $t_{ib}$ after receiving the broadcast signal from the CR. The signal contains all the recorded timestamps $T_{is}^{d}$, $T_{is}$, $T_{ih}$ and $T_{ib}$ of itself. The CR records the arrival time of the signal as $T_{hi}$. The measurement models of the signals from the NRs to CR are represented as(15)$$\begin{array}{cc} T_{ib} & =\mu_{i}t_{ib}+o_{i}, \end{array}$$
(16)$$\begin{array}{cc} T_{hi} & =\mu_{h}\left(t_{ib}+\tau_{hi}\right)+o_{h}+n_{hi},i=1,2,\cdots ,M, \end{array}$$where $n_{hi}∼N\left(0,\sigma_{h}^{2}\right),i=1,2,\cdots ,M$ are independent and identically distributed (IID) Gaussian measurement noises.

## 3. The UTLS Algorithm

We considered the problem of underwater target localization and synchronization for the distributed SIMO sonar system. Firstly, we combined the direct signals and the broadcast signals by the receivers to carry out time synchronization. Sequentially, we localized the target with the estimated clock parameters and the TOAs of reflected signals.

### 3.1. Time Synchronization

In this section, we perform time synchronization by estimating the clock skews of the receivers firstly, and estimating the clock offsets of the receivers sequentially.

#### 3.1.1. Estimation of the Clock Skews

Different from the situation of our previous work [4], there are uncertainties in the locations of the receivers. In order to reduce the influence of the uncertainties on the target localization, we estimate the clock skews of the receivers in conjunction with the locations of the receivers with EM algorithm. By combining (Equation 13)–(Equation 16), we have(17a)$$\begin{array}{cc} T_{hb} & =\mu_{h}\left(\frac{T_{ih}}{\mu_{i}}-\tau_{hi}\right)-\frac{\mu_{h}o_{i}}{\mu_{i}}+o_{h}-\frac{\mu_{h}}{\mu_{i}}n_{ih}, \end{array}$$
(17b)$$\begin{array}{cc} T_{hi} & =\mu_{h}\left(\frac{T_{ib}}{\mu_{i}}+\tau_{hi}\right)-\frac{\mu_{h}o_{i}}{\mu_{i}}+o_{h}+n_{hi},\\ i & =1,2,\cdots ,M, \end{array}$$which give that(18)$$\begin{array}{c} T_{hi}-T_{hb}=\mu_{h}\left(\frac{T_{ib}-T_{ih}}{\mu_{i}}+2\tau_{hi}\right)+{\tilde{\varepsilon }}_{i}, \end{array}$$where ${\tilde{\varepsilon }}_{i}=\frac{\mu_{h}}{\mu_{i}}n_{ih}+n_{hi}$ denoting the noise. It follows from (Equation 11) and ([Disp-formula FD17b-sensors-19-01976]) that(19a)$$\begin{array}{cc} \frac{T_{hi}^{d}-o_{h}-n_{h}^{d}}{\mu_{h}}-\frac{T_{is}^{d}-o_{i}-n_{i}^{d}}{\mu_{i}} & =\tau_{h}^{d}-\tau_{i}^{d}, \end{array}$$
(19b)$$\begin{array}{cc} \frac{T_{hi}-o_{h}-n_{hi}}{\mu_{h}}-\frac{T_{ib}-o_{i}}{\mu_{i}} & =\tau_{hi}, \end{array}$$which can be combined as(20)$$\begin{array}{c} T_{hi}-T_{hs}^{d}=\mu_{h}\left(\frac{T_{ib}-T_{is}^{d}}{\mu_{i}}+\tau_{hi}-\tau_{h}^{d}+\tau_{i}^{d}\right)+{\bar{\varepsilon }}_{i}, \end{array}$$where ${\bar{\varepsilon }}_{i}=\frac{\mu_{h}n_{i}^{d}}{\mu_{i}}+n_{hi}-n_{h}^{d}$ denoting the noise. We combine (18) and (20) and represent it in the vector form as(21)$$\begin{array}{c} \varphi =H\Theta +\varepsilon , \end{array}$$where $\Theta ={\left[\vartheta^{T},\mu_{h}\right]}^{T}$ denoting the parameter vector, the vector $\vartheta ={\left[\mu_{h}/\mu_{1},\cdots ,\mu_{h}/\mu_{M}\right]}^{T}$, and the vector $\varepsilon ={\left[{\bar{\varepsilon }}_{1},\cdots ,{\bar{\varepsilon }}_{M},{\tilde{\varepsilon }}_{1},\cdots ,{\tilde{\varepsilon }}_{M}\right]}^{T}$ denoting the noises. The measurements related to the target position are not included in (21). Therefore, (21) is independent on the target position. The covariance matrix of ε is of the form(22)$$\begin{array}{c} \Sigma_{\varepsilon }=\left[\begin{array}{cc} \Sigma_{11} & \Sigma_{12}\\ \Sigma_{21} & \Sigma_{22} \end{array}\right], \end{array}$$where $\Sigma_{22}=\mathrm{diag}\left(\sigma_{h}^{2}+\mu_{h}^{2}\sigma_{1}^{2}/\mu_{1}^{2},\cdots ,\sigma_{h}^{2}+\mu_{h}^{2}\sigma_{M}^{2}/\mu_{M}^{2}\right)$, $\Sigma_{11}=\Sigma_{22}+\sigma_{h}^{2}1_{M}1_{M}^{T}$, $\Sigma_{12}=\Sigma_{21}=\sigma_{h}^{2}I_{M}$. The vector $\varphi \in R^{2M}$ is the observation which can be stated as$$\begin{array}{cc} \varphi & ={\left[T_{h1}-T_{hs}^{d},\cdots ,T_{hM}-T_{hs}^{d},T_{h1}-T_{hb},\cdots ,T_{hM}-T_{hb}\right]}^{T}. \end{array}$$
The matrix $H=\left[A\;h\right]\in R^{2M\times (M+1)}$ denoting the observation coefficient matrix, where $A={\left[A_{1}^{T}\;A_{2}^{T}\right]}^{T}$, $A_{1}=\mathrm{diag}\left(T_{ib}-T_{is}^{d},\cdots ,T_{Mb}-T_{Ms}^{d}\right)$, $A_{2}=\mathrm{diag}\left(T_{1b}-T_{1h},\cdots ,T_{Mb}-T_{Mh}\right)$ and $h={\left[\tau_{h1}-\tau_{h}^{d}+\tau_{1}^{d},\cdots ,\tau_{hM}-\tau_{h}^{d}+\tau_{M}^{d},2\tau_{h1},\cdots ,2\tau_{hM}\right]}^{T}$. The entries of h are related to the locations of the receivers which are not exactly known. In order to obtain a fixed observation matrix, we linearize the entries of h by the Taylor expansion as$$\begin{array}{cc} \tau_{hi} & \approx {\bar{\tau }}_{hi}+\frac{\partial {\bar{\tau }}_{hi}}{\partial x_{i}}\left(x_{i}-{\bar{x}}_{i}\right)+\frac{\partial {\bar{\tau }}_{hi}}{\partial x_{h}}\left(x_{h}-{\bar{x}}_{h}\right),\\ \tau_{h}^{d} & \approx {\bar{\tau }}_{h}^{d}+\frac{\partial {\bar{\tau }}_{h}^{d}}{\partial x_{h}}\left(x_{h}-{\bar{x}}_{h}\right),\\ \tau_{i}^{d} & \approx {\bar{\tau }}_{i}^{d}+\frac{\partial {\bar{\tau }}_{i}^{d}}{\partial x_{h}}\left(x_{i}-{\bar{x}}_{i}\right), \end{array}$$where the vectors ${\bar{x}}_{h}$ and ${\bar{x}}_{i}$ are the Taylor expansion points of the propagation delays $\tau_{h}^{d}$ and $\tau_{i}^{d}$, respectively. The vector ${\left[{\bar{x}}_{h}^{T}\;{\bar{x}}_{i}\right]}^{T}$ is the Taylor points of the delay $\tau_{hi}$. The propagation delays ${\bar{\tau }}_{hi}=f_{hi}\left({\bar{x}}_{h},{\bar{x}}_{i}\right)$, ${\bar{\tau }}_{h}^{d}=f_{hs}\left({\bar{x}}_{h}\right)$ and ${\bar{\tau }}_{i}^{d}=f_{is}\left({\bar{x}}_{i}\right)$. The vector h can be approximated as(23)$$\begin{array}{c} h\approx \bar{h}+B\gamma , \end{array}$$where $\bar{h}={\left[{\bar{\tau }}_{h1}-{\bar{\tau }}_{h}^{d}+{\bar{\tau }}_{1}^{d},\cdots ,{\bar{\tau }}_{hM}-{\bar{\tau }}_{h}^{d}+{\bar{\tau }}_{M}^{d},2{\bar{\tau }}_{h1},\cdots ,2{\bar{\tau }}_{hM}\right]}^{T}$ denoting the Taylor expansion point of h. The vector γ is named as latent vector whose entries are latent variables. It can be expressed as(24)$$\begin{array}{cc} \gamma & =x_{r}-{\bar{x}}_{r}, \end{array}$$where$$\begin{array}{cc} x_{r} & ={\left[x_{1}^{T}\;\cdots \;x_{M}^{T}\;x_{h}^{T}\right]}^{T}\\ {\bar{x}}_{r} & ={\left[{\bar{x}}_{1}^{T}\;\cdots \;{\bar{x}}_{M}^{T}\;{\bar{x}}_{h}^{T}\right]}^{T}. \end{array}$$

The matrix $B={\left[B_{1}^{T}\;B_{2}^{T}\right]}^{T}\in R^{2M\times 3(M+1)}$ collecting the partial derivatives of h w.r.t. the locations of the receivers. Its submatrices can be expressed as$$\begin{array}{c} B_{1}=\left[\begin{array}{cccc} {\left(\frac{\partial {\bar{\tau }}_{h1}^{T}}{\partial x_{1}}+\frac{\partial {\bar{\tau }}_{1}^{d}}{\partial x_{1}}\right)}^{T} & \cdots & 0 & {\left(\frac{\partial {\bar{\tau }}_{h1}}{\partial x_{h}}-\frac{\partial {\bar{\tau }}_{h}^{d}}{\partial x_{h}}\right)}^{T}\\ \vdots & \ddots & \vdots & \vdots \\ 0 & \cdots & {\left(\frac{\partial {\bar{\tau }}_{hM}}{\partial x_{M}}+\frac{\partial {\bar{\tau }}_{M}^{d}}{\partial x_{M}}\right)}^{T} & {\left(\frac{\partial {\bar{\tau }}_{hM}}{\partial x_{h}}-\frac{\partial {\bar{\tau }}_{h}^{d}}{\partial x_{h}}\right)}^{T} \end{array}\right] \end{array}$$
$$\begin{array}{c} B_{2}=\left[\begin{array}{cccc} 2\frac{\partial {\bar{\tau }}_{h1}^{T}}{\partial x_{1}} & \cdots & 0 & 2\frac{\partial {\bar{\tau }}_{h1}^{T}}{\partial x_{h}}\\ \vdots & \ddots & \vdots & \vdots \\ 0 & 0 & 2\frac{\partial {\bar{\tau }}_{hM}^{T}}{\partial x_{M}} & 2\frac{\partial {\bar{\tau }}_{hM}^{T}}{\partial x_{h}} \end{array}\right]. \end{array}$$

By substituting (23) into (21), we can modify (21) to(25)$$\begin{array}{c} \varphi \approx \bar{H}\Theta +\mu_{h}B\gamma +\varepsilon , \end{array}$$where $\bar{H}=\left[A\;\bar{h}\right]$, the set {φ,γ} is termed complete data set. The parameter Θ can be estimated if the complete data set is known. The EM algorithm can be utilized to estimate the latent vector γ firstly and the parameter vector Θ sequentially.

We employ the EM algorithm to estimate the parameter vector Θ by maximizing the expectation of the complete data logarithm likelihood function under the posterior distribution of the latent variables. The process consists of two steps named as the expectation step (E-step) and the maximization step (M-step). The purpose of the E-step is to estimate the latent variables by calculating the conditional expectation. The M-step aims to estimate the parameters with the estimated latent variables. They are explained as follows.

E-Step:

This step involves estimating the latent variables γ and formulating the objective function which can be stated as(26)$$\begin{array}{c} Q\left(\Theta ,\Theta^{\left(l-1\right)}\right)=E\left[\log p\left(\varphi ,\gamma |\Theta \right)|\varphi ,\Theta^{\left(l-1\right)}\right], \end{array}$$where *l* is the iteration index, and logp(φ,γ|Θ) is the likelihood function which can be expressed as(27)$$\begin{array}{c} \log p(\varphi ,\gamma |\Theta )=\log p(\varphi |\gamma ,\Theta )+\log p(\gamma |\Theta ). \end{array}$$

Since the latent vector γ and the parameter vector Θ are independent, the second term in the right hand side of (27) can be omitted as it is independent on Θ. Therefore the function $Q(\Theta ,\Theta^{\left(l-1\right)})$ can be represented as(28)$$\begin{array}{c} Q\left(\Theta ,\Theta^{\left(l-1\right)}\right)\propto E\left[\log p\left(\varphi |\gamma ,\Theta \right)|\varphi ,\Theta^{\left(l-1\right)}\right]. \end{array}$$

Through the derivation (see Appendix A), we rewrite the objective function as(29)$$\begin{array}{cc} Q(\Theta ,\Theta^{\left(l-1\right)}) & \propto -\frac{1}{2}{\left[\varphi -A\vartheta -\bar{h}\mu_{h}\right]}^{T}{\Sigma_{\varepsilon }}^{-1}\left[\varphi -A\vartheta -\bar{h}\mu_{h}\right]\\ & +{\left[\varphi -A\vartheta -\bar{h}\mu_{h}\right]}^{T}\Sigma_{\varepsilon }^{-1}B\hat{\gamma }\mu_{h}-\frac{{\hat{V}}_{s}}{2}\mu_{h}^{2}, \end{array}$$where the variables $\hat{\gamma }$ and ${\hat{V}}_{s}$ can be calculated according to (A2) and (A4), respectively.

M-Step:

The maximization step attempts to find an optimal Θ which maximizes the objective function $Q(\Theta ,\Theta^{\left(l-1\right)})$. We take the first order derivative of the object function $Q(\Theta ,\Theta^{l-1})$ w.r.t. Θ and set the result to zero, which gives(30)$$\begin{array}{c} A^{T}\Sigma_{\varepsilon }^{-1}\left(\varphi -A\vartheta -\bar{h}\mu_{h}-B\hat{\gamma }\mu_{h}\right)=0,\\ \left[{\bar{h}}^{T}\Sigma_{\varepsilon }^{-1}+{\hat{\gamma }}^{T}B^{T}\Sigma_{\varepsilon }^{-1}\right)\left(\varphi -A\vartheta -\bar{h}\mu_{h}\right] \end{array}$$
(31)$$\begin{array}{c} -{\bar{h}}^{T}\Sigma_{\varepsilon }^{-1}B\hat{\gamma }\mu_{h}-{\hat{V}}_{s}\mu_{h}=0. \end{array}$$

Define(32)$$\begin{array}{c} q=\left[\begin{array}{c} A^{T}\Sigma_{\varepsilon }^{-1}\varphi \\ \left[{\bar{h}}^{T}+{\hat{\gamma }}^{T}B^{T}\right]\Sigma_{\varepsilon }^{-1}\varphi \end{array}\right], \end{array}$$
and
(33)$$\begin{array}{c} G=\left[\begin{array}{cc} G_{11} & G_{12}\\ G_{21} & G_{22} \end{array}\right], \end{array}$$where$$\begin{array}{cc} G_{11} & =A^{T}\Sigma_{\varepsilon }^{-1}\left(\bar{h}+B\hat{\gamma }\right),\\ G_{12} & =A^{T}\Sigma_{\varepsilon }^{-1}A,\\ G_{21} & ={\left(\bar{h}\right)}^{T}\Sigma_{\varepsilon }^{-1}\bar{h}+2{\left(\bar{h}\right)}^{T}\Sigma_{\varepsilon }^{-1}B\hat{\gamma }+{\hat{V}}_{s},\\ G_{22} & =[{\left(\bar{h}\right)}^{T}\Sigma_{\varepsilon }^{-1}+{\hat{\gamma }}^{T}B^{T}\Sigma_{\varepsilon }^{-1}]A. \end{array}$$

The Equations (Equation 30) and (Equation 31) can be combined into the following vector form(34)$$\begin{array}{c} G\Theta =q, \end{array}$$which gives the estimate of Θ as(35)$$\begin{array}{c} \Theta^{\left(l\right)}=G^{-1}q. \end{array}$$

According to (23), errors have been introduced by the linearization of h. In order to reduce the errors, we need to update $\bar{h}$ by updating the Taylor expansion point ${\bar{x}}_{r}$. Assuming that ${\bar{x}}_{r}^{\left(1\right)}={\overset{ˇ}{x}}_{r}$ during the first iteration, where ${\overset{ˇ}{x}}_{r}={\left[{\overset{ˇ}{x}}_{1}^{T}\;\cdots \;{\overset{ˇ}{x}}_{M}^{T}\;{\overset{ˇ}{x}}_{h}^{T}\right]}^{T}$. During the *l*th iteration, we set ${\bar{x}}_{r}^{\left(l\right)}={\hat{x}}_{r}^{\left(l-1\right)}+{\hat{\gamma }}^{\left(l-1\right)}$ to reduce the difference between ${\hat{x}}_{r}^{\left(l\right)}$ and the real locations $x_{r}$ of the receivers. Actually, during the *l*-th iteration, the latent vector ${\hat{\gamma }}^{\left(l\right)}$ collects the location offsets of the receivers, while ${\bar{x}}_{r}^{\left(l\right)}$ is the estimate of the receivers’ locations. Therefore, we can estimate the locations of the receivers after the last iteration by ${\hat{x}}_{r}={\bar{x}}_{r}^{\left(l\right)}+{\hat{\gamma }}^{\left(l\right)}$. Denote $\hat{\Theta }$ as the estimate of Θ, the clock skews of the receivers can be calculated as follows(36)$$\begin{array}{cc} {\hat{\mu }}_{h} & ={\left[\hat{\Theta }\right]}_{M+1} \end{array}$$
(37)$$\begin{array}{cc} {\hat{\mu }}_{i} & =\frac{{\hat{\mu }}_{h}}{{\left[\hat{\Theta }\right]}_{i}},i=1,2,\cdots ,M. \end{array}$$

The details of the EM based clock skews estimation algorithm are stated in Algorithm 1.

**Algorithm 1**: Expectation maximization (EM) based clock skews estimation algorithm.**Require:**Tib,Thi,Tih,i=1,2,⋯,M, Thb, a,b, xh, L,Nxˇi,Tisd,i=1,2,⋯,M,h,**Ensure:**μ^i,i=1,2,⋯,M,h, x^r 1: l=1, index=1, x^r(l−1)=[xˇ1T,⋯,xˇh]T 2: **while**index**do** 3: x¯r(l)=xr(l−1) 4: linearize h at x¯r according to (23) 5: **E-Step:** 6:  calculate γ^(l) according to (A2) 7:  calculate V^s(l) according to (A4) 8: **M-Step:** 9:  calculate Θ(l) according to (35)10:   **if**
|Θ(l)−Θ(l−1)|≤δ or
l≥L
**then**11:    Θ^=Θ(l), x^r=x¯r(l)+γ^(l), index=012:   **else**13:    Θ(l−1)=Θ(l), x^r(l−1)=x¯r(l)+γ^(l), l=l+114:   **end if**15: **end while**16: μ^h=[Θ^]M+1, μ^i=μ^h/[Θ^]i,i=1,2,⋯,M.17: **return**
μ^i,i=1,2,⋯,M,h, x^r.

We remark that the uncertainties of the receiver locations are considered in two steps. In the first step, the statistical properties of the receiver locations are utilized to estimate the location offsets and obtain the estimate $\hat{\gamma }$. In the second step, the estimate $\hat{\gamma }$ is used to update the receiver locations, which can reduces the influence of the uncertainties of the receiver locations.

#### 3.1.2. Estimation of the Clock Offsets

We employ the WLS algorithm to estimate the clock offsets of the receivers. By substituting the estimates of the clock skews into (Equation 11), ([Disp-formula FD17a-sensors-19-01976]) and ([Disp-formula FD17b-sensors-19-01976]), we have$$\begin{array}{c} \varphi_{o}=H_{o}o+\epsilon , \end{array}$$where the vector $\varphi_{o}\in R^{3M+1}$ can be stated as$$\begin{array}{c} {}[T_{1s}^{d}-{\hat{\mu }}_{1}\left(t_{0}+{\hat{\tau }}_{i}^{d}\right),\cdots ,T_{hs}^{d}-{\hat{\mu }}_{h}\left(t_{0}+{\hat{\tau }}_{hs}^{d}\right),\\ T_{1h}-{\hat{\mu }}_{1}\left(T_{hb}/{\hat{\mu }}_{h}+{\hat{\tau }}_{h1}\right),\cdots ,T_{Mh}-{\hat{\mu }}_{M}\left(T_{hb}/{\hat{\mu }}_{h}+{\hat{\tau }}_{hM}\right),\\ T_{h1}-{\hat{\mu }}_{h}\left(T_{1b}/{\hat{\mu }}_{1}+{\hat{\tau }}_{h1}\right),\cdots ,T_{hM}-{\hat{\mu }}_{h}\left(T_{Mb}/{\hat{\mu }}_{M}+{\hat{\tau }}_{hM}\right){]}^{T}, \end{array}$$the delay ${\hat{\tau }}_{i}^{d}=\tau_{i}^{d}\left({\hat{x}}_{i}\right)$, ${\hat{\tau }}_{hi}=\tau_{hi}\left({\hat{x}}_{h},{\hat{x}}_{i}\right)$. The coefficient matrix is defined as $H_{o}={\left[H_{o1}^{T}\;H_{o2}^{T}\;H_{o3}^{T}\right]}^{T}\in R^{\left(3M+1)\times (M+1\right)}$, where $H_{o1}=I_{M+1}$, $H_{o2}=\left[I_{M},{\left[-\mu_{1}/\mu_{h},\cdots ,1\mu_{M}/\mu_{h}\right]}^{T}\right]$ and $H_{o3}=\left[-\mathrm{diag}\left(\mu_{h}/\mu_{1},\cdots ,\mu_{h}/\mu_{M}\right),1_{M}\right]$. The vector $\epsilon ={\left[n_{1}^{d},\cdots ,n_{M}^{d},n_{h}^{d},n_{1h},\cdots ,n_{Mh},n_{h1},\cdots ,n_{hM}\right]}^{T}\in R^{3M+1}$ denoting the noise. Its covariance matrix can be expressed as $\Sigma_{\epsilon }=\mathrm{diag}\left(\Sigma_{\epsilon 1},\Sigma_{\epsilon 2},\sigma_{h}^{2}I_{M}\right)$, where $\Sigma_{\epsilon 1}=\mathrm{diag}\left(\sigma_{1}^{2},\cdots ,\sigma_{M}^{2},\sigma_{h}^{2}\right)$, $\Sigma_{\epsilon 2}=\mathrm{diag}\left(\sigma_{1}^{2},\cdots ,\sigma_{M}^{2}\right)$.

By applying the WLS algorithm, we can express the estimates of the clock offsets as(38)$$\begin{array}{c} \hat{o}={\left(H_{o}^{T}W_{o}H_{o}\right)}^{-1}H_{o}^{T}W_{o}\varphi_{o}, \end{array}$$where the weighted matrix is defined as $W_{o}=\Sigma_{\epsilon }^{-1}$.

### 3.2. Target Localization

By utilizing the estimates of the clock parameters, (12) can be rewritten as(39)$$\begin{array}{c} \phi_{i}=f_{s}\left(x,x_{s}\right)+f_{i}\left(x,x_{i}\right)+\frac{n_{i}}{{\hat{\mu }}_{i}},i=1,2,\cdots ,M,h, \end{array}$$where $\phi_{i}=\left(T_{is}-{\hat{o}}_{i}-{\hat{\mu }}_{i}t_{0}\right)/\left({\hat{\mu }}_{i}\right)$. Define $\phi ={\left[\phi_{1},\cdots ,\phi_{M},\phi_{h}\right]}^{T}$, $f\left(x,x_{r}\right)={\left[f_{1}\left(x,x_{1}\right),\cdots ,f_{M}\left(x,x_{M}\right),f_{h}\left(x,x_{h}\right)\right]}^{T}$ and $\omega ={\left[n_{1}/{\hat{\mu }}_{i},\cdots ,n_{M}/{\hat{\mu }}_{M},n_{h}/{\hat{\mu }}_{h}\right]}^{T}$ as the observation vector, the propagation delay vector and the noise vector, respectively. We rewrite (39) as(40)$$\begin{array}{c} \phi =f_{s}\left(x,x_{s}\right)1_{M+1}+f\left(x,x_{r}\right)+\omega . \end{array}$$

The covariance matrix of the noise ω is $\Sigma_{\omega }=\mathrm{diag}\left(\sigma_{1}^{2}/\mu_{1}^{2},\cdots ,\sigma_{M}^{2}/\mu_{M}^{2},\sigma_{h}^{2}/\mu_{h}^{2}\right)\in R^{\left(M+1)\times (M+1\right)}$. It is hard to give an analytical solution of the estimation of the target’s location as the propagation delays $f_{s}\left(x,x_{s}\right)$ and $f_{i}\left(x,x_{i}\right)$ are nonlinear w.r.t. the variable x. We employ the NWLS algorithm for the target localization. The iterative process can be expressed as(41)$$\begin{array}{c} x^{\left(l\right)}=x^{\left(l-1\right)}+{\left[{\left(D^{\left(l-1\right)}\right)}^{T}WD^{\left(l-1\right)}\right]}^{-1}{\left(D^{\left(l-1\right)}\right)}^{T}W\\ \times [\phi -f_{s}\left(x^{\left(l-1\right)},x_{s}\right)-f\left(x^{\left(l-1\right)},{\hat{x}}_{r}\right)], \end{array}$$where the weight matrix $W=\Sigma_{\omega }^{-1}$, and the matrix $D^{\left(l-1\right)}$ is the Jacobi matrix of $f_{s}\left(x,x_{s}\right)1_{M+1}+f\left(x,{\hat{x}}_{r}\right)$ w.r.t. x. For the *l*-th iteration, the Jacobi matrix $D^{\left(l-1\right)}$ can be stated as(42)$$\begin{array}{c} D^{\left(l-1\right)}=\left[\frac{df_{s}\left(x,x_{s}\right)}{dx^{T}}⊗1_{M+1}+\frac{df(x,{\hat{x}}_{r})}{dx^{T}}\right]{|}_{x=x^{\left(l-1\right)}}. \end{array}$$

## 4. Numerical Simulations

We evaluate the performance of our proposed algorithms via the software “Matlab” with an isogradient SSP and different variances of the locations of the receivers. At first, we introduce the simulation setup. Then we show the estimation performance including the parameters of the receivers and the location of the target.

### 4.1. Simulation Setup

As the clock skews of the receivers are designed to be 1, we assume that the real value of the clock skews follow the Gaussian distributions $\mu_{i}∼N\left(1,\sigma_{\mu_{i}}^{2}\right),i=1,2,\cdots ,M,h$, where $\sigma_{\mu_{i}}^{2}$ is the variance of $\mu_{i}$. Assume that $\sigma_{\mu_{i}}^{2}=\sigma_{\mu_{j}}^{2}\in \left(10^{-12},10^{-8}\right]$ and $\sigma_{i}^{2}=\sigma_{j}^{2}\in \left(10^{-8},10^{-6}\right]$ for i,j=1,2,⋯,M,h. The receivers are random anchored in the rectangle [−800 m, 800 m] × [−800 m, 800 m] at the bottom of the sea in a star topology with depths around 3000 m. The anchor chain length of the receivers are limited in the interval (0 m, 300 m]. We assume that the measurement variances of the receiver locations are within the range of [−10dB,8dB] in dB. The location of the target is a three-dimensional random variable which is uniform distributed in the cube [−800 m, 800 m] × [−800 m, 800 m] × [200 m, 3000 m]. We assume that the parameters of the sound speed as a=0.016,b=1500. In order to fit the statistical properties of the noise samples, 2000 Monte Carlo trails are utilized to evaluate the performance of our proposed algorithms by comparing with the competitors listed in Table 1.

As shown in Table 1, the competitors include the NWLS-UTL algorithm, the WLS-NWLS algorithm, the AWLS-NWLS algorithm, the ANWLS algorithm, and the NWLS-RP algorithm. The NWLS-UTL algorithm is proposed in [4] for underwater target localization. In our proposed UTLS algorithm, the NWLS algorithm is employed to estimate the location of the target with the values of the clock skews and clock offsets estimated by the EM and WLS algorithms, respectively. For comparison, the WLS-NWLS algorithm employs the NWLS algorithm to localize the target with the estimates of the clock skews and clock offsets obtained by the WLS algorithms. The difference between the AWLS-NWLS and WLS-NWLS is that the values of the clock skews, which are utilized in the estimations of the clock offsets and target location, are estimated by an AWLS algorithm in the AWLS-NWLS algorithm. In the ANWLS algorithm, the clock skews and clock offsets of the receivers are assumed to be “1” and “0”, respectively. The NWLS-RP algorithm acts as a benchmark since the real values of the clock parameters and the locations of the receivers are used. While the measurements of the receiver locations are used in the NWLS-UTL, the WLS-NWLS, the AWLS-NWLS and ANWLS algorithms.

Firstly, we illustrate the necessity of the consideration of the uncertainties of the receiver locations. We show the performance of the estimation of the clock skews by the comparison of the proposed EM algorithm with the WLS algorithm of the WLS-NWLS algorithm, the AWLS algorithm of the AWLS-NWLS algorithm (whose details are shown in Appendix B) and the tailored WLS of the NWLS-UTL algorithm. We show the performance of the estimations of the clock offsets by comparing the WLS algorithms of the UTLS, the WLS-NWLS and the AWLS-NWLS algorithms. Sequently, we show the necessarity of taking time synchronization into account. We reveal the target localization performance of our proposed UTLS algorithm by comparing the NWLS algorithm of the UTLS with the NWLS algorithms of the WLS-NWLS, the AWLS-NWLS, the ANWLS and the NWLS-RP algorithms. In addition, we compare the UTLS algorithm with the NWLS-UTL algorithm which is proposed in our previous work [4] to show the improvement.

For ease of presentation, we replace the root mean square errors (RMSEs) of the estimations of the clock parameters and the locations of the NRs with the corresponding average value which can be stated as(43)$$\begin{array}{c} ave\left(\mathrm{RMSE}\right)=\frac{1}{N}\sum_{i=1}^{N}\mathrm{RMSE}\left(variable_{i}\right), \end{array}$$where the $variable_{i}$ can be chosen as $\mu_{i}$, $o_{i}$ or $x_{i}$, the variable *N* denotes the number of the $variable_{i}$. For example, N=M if the $variable_{i}$ stands for $\mu_{i}$, i=1,2,⋯,M. For simplicity, we utilize “the RMSE of $\mu_{i}\backslash o_{i}\backslash x_{i}$” to stand for “the average RMSE of $\mu_{i}\backslash o_{i}\backslash x_{i},i=1,2,\cdots ,N$”, from now on.

### 4.2. Estimation Performance

The simulation results are shown in Figure 5, Figure 6, Figure 7, Figure 8, Figure 9, Figure 10 and Figure 11. In Figure 5, Figure 8 and Figure 10, we fix $\sigma_{\mu_{h}}^{2},\sigma_{\mu_{i}}^{2},\sigma_{x}^{2}$ and limit the term $10\log(1/\sigma_{i}^{2})$ in the range [60dB,78dB]. While in Figure 6, Figure 9 and Figure 11, we fix $\sigma_{\mu_{h}}^{2},\sigma_{\mu_{i}}^{2},\sigma_{i}^{2},\sigma_{h}^{2}$ and limit the term $10\log(1/\sigma_{x}^{2})$ in the range [−10dB,8dB].

#### 4.2.1. Performance of the Estimations of the Clock Skews and Locations of the Receivers

In this subsection, we show the performance of the clock skew estimations carried out by the algorithms listed in the second column of Table 1. The estimation performance of the receiver locations via the EM algorithm of the UTLS algorithm is shown in this subsection too. The simulation results of the estimation of the clock skews and the receiver locations are shown in Figure 5, Figure 6 and Figure 7. For the estimation performance of the clock skews and the receiver locations, we have the following remarks

The UTLS algorithm is the best one, the WLS-NWLS algorithm is the second one and the AWLS-NWLS is the worst one.The performance of the estimation of the clock skews can be improved by updating the locations of the receivers. The main difference between the EM algorithm of the UTLS algorithm and the WLS algorithm of the WLS-NWLS algorithm is that the EM algorithm estimates the locations of the receivers by calculating the conditional expectation and utilizes the estimates of the locations for the estimation of the clock skews, while the WLS algorithm of the WLS-NWLS algorithm only utilizes the statistical properties for the estimation of the clock skews. As shown in Figure 5, the UTLS algorithm is better than the WLS-NWLS algorithm and the difference increases with the decreasing of the variance $\sigma_{i}^{2}$. This is because that with a smaller variance $\sigma_{i}^{2}$, the UTLS algorithm obtains more accurate estimates of the locations of the receivers as shown in the upper subplot of Figure 7, with which the estimation accuracy of the clock skews can be improved.By utilizing the statistical properties of the locations of the receivers, the estimation performance of clock skews can be improved as shown in Figure 5. The main difference between the WLS algorithm of the WLS-NWLS algorithm and the AWLS algorithm of the AWLS-NWLS algorithm is that the WLS algorithm utilizes the statistical properties of the receivers’ locations, while the AWLS dose not. The performance superiority of the WLS-NWLS algorithm compared with the AWLS-NWLS algorithm arises from the utilizing of the statistical properties.In the NWLS-UTL algorithm, the clock offsets are eliminated by a designed subtraction of the time delay. The clock skews are estimated by a tailored WLS algorithm. Its. estimation performance of the clock skews is similar to the WLS-NWLS and the AWLS-NWLS algorithms and poorer than the UTLS algorithm.As shown in Figure 6, the larger the variance $\sigma_{x}^{2}$ of the receiver location noises, the greater the superiority of the UTLS algorithm.The UTLS algorithm is more robust against the location uncertainties of the receivers than the WLS-NWLS, the AWLS-NWLS and the NWLS-UTL algorithms. As shown in Figure 6 the performance of the UTLS algorithm decreases slowly with the increasing of the variance $\sigma_{x}^{2}$. While the WLS-NWLS, the AWLS-NWLS and the NWLS-UTL algorithms decrease faster than the UTLS algorithm with the increasing of the variance $\sigma_{x}^{2}$. The main reason is that the accuracy of the estimates of the receivers’ locations via the EM algorithm of the UTLS algorithm declines slowly with the increasing of the variance $\sigma_{x}^{2}$ as shown in the lower subplot of Figure 7.

#### 4.2.2. Performance of the Estimations of the Clock offsets of the Receivers

In this subsection, we show the performance of the clock offset estimations carried out by the algorithms listed in the third column of Table 1. We show the simulation results of the estimation of the clock offsets in Figure 8 and Figure 9. For the estimate of the clock offsets, we have the following remarks

For the estimation of the offsets of the receivers, the UTLS algorithm is the best and the differences between the UTLS algorithm and the other two algorithms increase with the decreasing of the variance $\sigma_{i}^{2}$ as shown in Figure 8. There are two reasons. The first one is that the differences of the clock skew estimates between the EM algorithm of the UTLS algorithm and the other two algorithms increase with the decreasing of the variance $\sigma_{i}^{2}$. The second one is that the EM algorithm of the UTLS algorithm provides more accurate locations of the receivers for the WLS algorithm of the UTLS algorithm, and the estimation accuracy of the locations of the receivers increases with the decreasing of the variance $\sigma_{i}^{2}$.The performance superiority of the WLS algorithm of the WLS-NWLS algorithm compared with the WLS algorithm of the AWLS-NWLS algorithm arises from the utilizing of the statistical properties of the receivers’ locations as shown in Figure 8.As shown in Figure 9, the WLS algorithm of the UTLS algorithm is more robust against the uncertainty of the receivers’ locations. This characteristic is inherited from the EM algorithm of the UTLS algorithm.

#### 4.2.3. Performance of the Estimation of the Location of the Target

In this subsection, we show the performance of the target localization carried out by the algorithms listed in the last column of Table 1. The simulation results of the target localization are shown in Figure 10 and Figure 11. We have the following remarks on the simulation results of the target localization.

The performance of the UTLS is the best compared with that of the WLS-NWLS, the AWLS-NWLS and the ANWLS algorithms as shown in Figure 10. The advantages of the UTLS arise from two reasons. First, the estimation accuracies of the clock skews and offsets carried out by the EM and WLS algorithms of the UTLS algorithm, respectively, are higher than that of the WLS-NWLS, the AWLS-NWLS and the ANWLS algorithms. Second, the estimation accuracy of the receivers’ locations by the EM algorithm of the UTLS algorithm is higher than the corresponding measurements of the receivers’ locations.The performance of the WLS-NWLS algorithm is better than the AWLS-NWLS and the ANWLS algorithms. This superiority of the WLS-NWLS arises from the WLS algorithms of the WLS-NWLS algorithm.As shown in Figure 10, the performance variation trend of the UTLS is almost the same as the NWLS-RP in which the real clock parameters and locations of the receivers are used. While the performance of the other algorithms almost do not vary with the variance of $\sigma_{i}^{2}$ when the variance $\sigma_{i}^{2}$ is small.The performance of the UTLS algorithm is almost the same as the WLS-NWLS algorithm when the variance $\sigma_{x}^{2}$ is small as shown in Figure 11. The reasons can be stated as follows. As shown in the lower subplot of Figure 7, the difference between the estimated locations of the receivers and the measurements of the locations is small when the variance $\sigma_{x}^{2}$ is small. As shown in Figure 6 and Figure 9, this phenomenon makes the performance superiority of the EM and WLS algorithms of our proposed UTLS algorithm small when compared to the WLS algorithms of the WLS-NWLS algorithm.The UTLS algorithm is still better than the AWLS-NWLS and ANWLS algorithms when the variance $\sigma_{x}^{2}$ is small. The larger the variance $\sigma_{x}^{2}$, the greater the superiority of the UTLS algorithm.The localization performance of the NWLS-UTL algorithm is poor as shown in Figure 10. This phenomenon may arise from the fact that the NWLS-UTL algorithm is susceptible to the location uncertainties of the receivers as shown Figure 11. The clock offsets have been eliminated by the NWLS-UTL algorithm. Therefore, the computational cost is smaller because of the absence of the clock offset estimates.The localization performance of the NWLS-RP algorithm is better the our proposed UTLS algorithm. The reason is that the real clock parameters and receiver locations are employed in the NWLS-RP algorithm, which is impractical.

In summary, the superiority of the EM algorithm of our proposed UTLS algorithm arises from the consideration of the uncertainties of the locations of the receivers and the estimation of the receivers’ locations. With the estimates of the EM algorithm, the WLS algorithm of our proposed UTLS algorithm obtains more accurate clock offsets. The NWLS algorithm of our proposed UTLS algorithm achieves more accurate target localization with the estimates of the clock skews and offsets obtained via the EM and WLS algorithms of our proposed UTLS algorithm, respectively. The superiority of the UTLS algorithm increase with the increasing of the variance $\sigma_{x}^{2}$.

## 5. Conclusions

In this work, we propose the target localization and time synchronization (UTLS) algorithm for the distributed SIMO sonar system with an isogradient SSP and the uncertainties of the locations of the receivers. The UTLS algorithm contains two steps. In the first step, we carry out time synchronization via the EM and the WLS algorithms of our proposed UTLS algorithm. The locations of the receivers are also estimated by the EM algorithm in this step. In the second step, the NWLS algorithm of our proposed UTLS algorithm is employed to perform target localization with the estimates of the clock parameters and the locations of the receivers. We compared the UTLS algorithm with the NWLS-UTL, the WLS-NWLS, the AWLS-NWLS, the ANWLS and the NWLS-RP algorithms. The simulation results show that our proposed UTLS algorithm is better than the NWLS-UTL, the WLS-NWLS, the AWLS-NWLS and the ANWLS algorithms, and the advantage of the UTLS increases with the increasing of the variance $\sigma_{x}^{2}$. An isogradient SSP is an approximation of the real underwater SSP. In order to obtain a more accurate measurement of distance, we will take a more elaborate SSP into consideration for target localization in our future work.

## Figures and Tables

**Figure 1 sensors-19-01976-f001:**
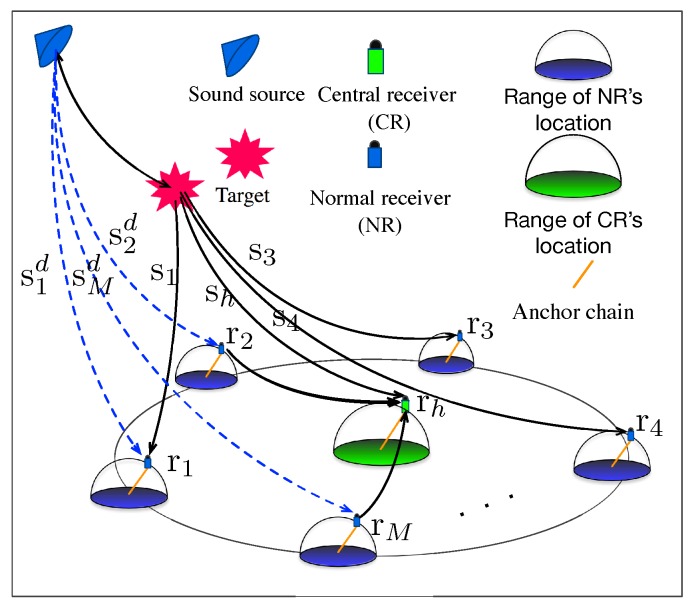
The schematic diagram of target localization by single-input multiple-output (SIMO) sonar systems.

**Figure 2 sensors-19-01976-f002:**
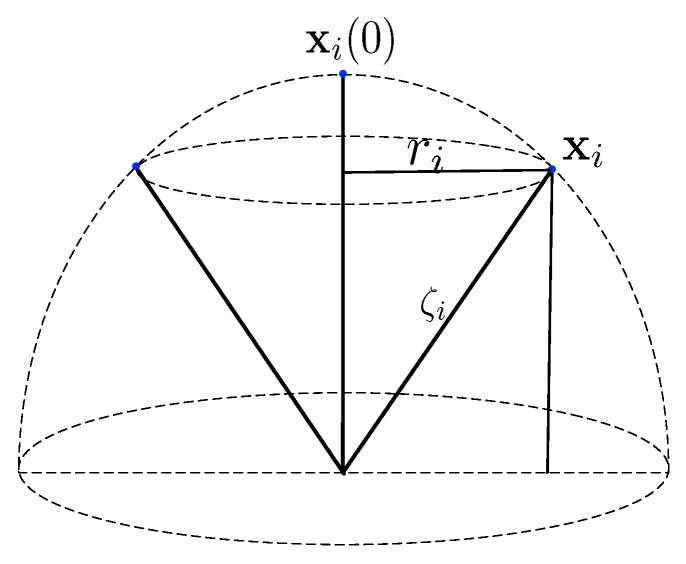
The feasible locations of the receivers.

**Figure 3 sensors-19-01976-f003:**
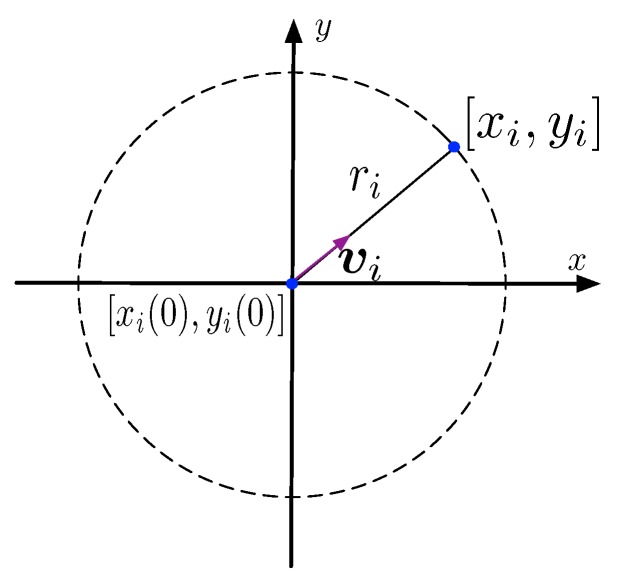
The scheme to calculate the coordinate $\left[x_{i},y_{i}\right]$.

**Figure 4 sensors-19-01976-f004:**
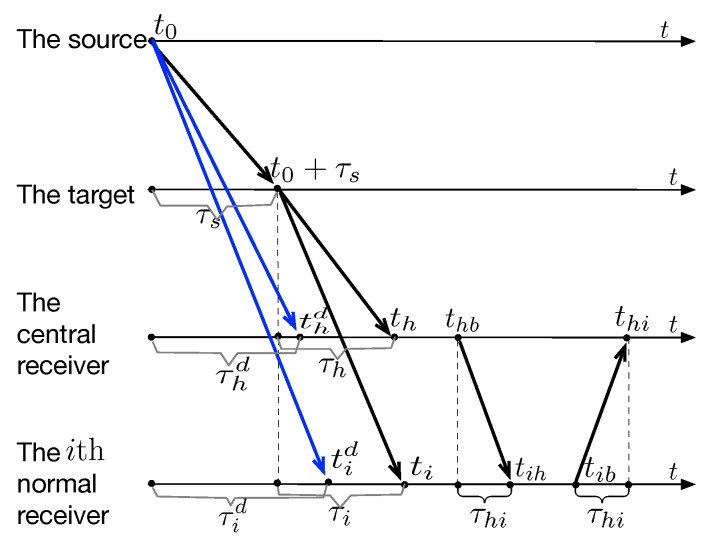
Description of the time series.

**Figure 5 sensors-19-01976-f005:**
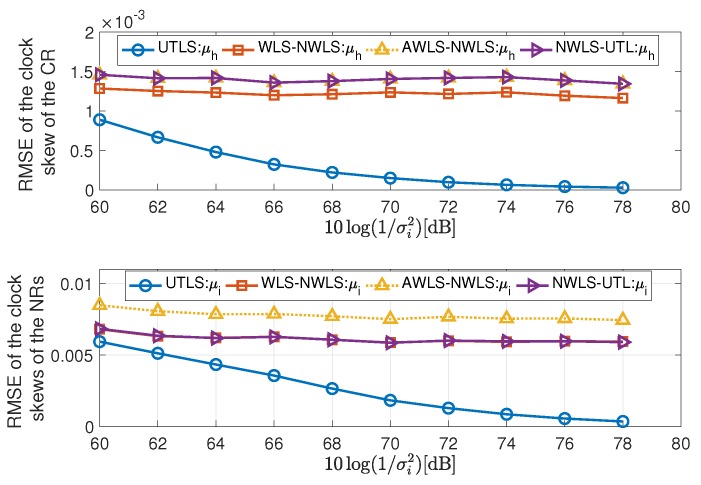
Root mean square errors (RMSEs) of the estimation of the clock skews of the receivers via the underwater target localization and synchronization (UTLS), the weighted least squares (WLS)-nonlinear weighted least squares (NWLS), the approximated WLS (AWLS)-NWLS and the NWLS-UTL algorithms, respectively, with $10\log\left(1/\sigma_{\mu_{h}}^{2}\right)=10\log\left(1/\sigma_{\mu_{i}}^{2}\right)=96$ dB and $10\log(1/\sigma_{x}^{2})=-10$ dB.

**Figure 6 sensors-19-01976-f006:**
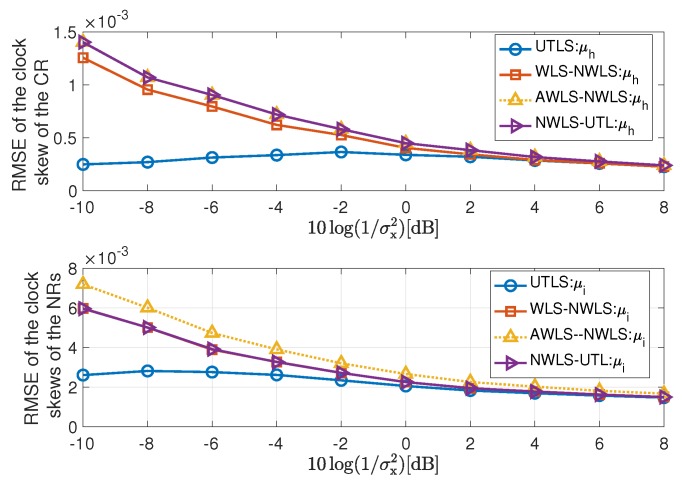
RMSEs of the estimation of the clock skews of the receivers via the UTLS, the WLS-NWLS, the AWLS-NWLS and the NWLS-underwater target localization (UTL) algorithms, respectively, with $10\log\left(1/\sigma_{\mu_{h}}^{2}\right)=10\log\left(1/\sigma_{\mu_{i}}^{2}\right)=80$ dB, and $10\log\left(1/\sigma_{i}^{2}\right)=10\log\left(1/\sigma_{h}^{2}\right)=68$ dB.

**Figure 7 sensors-19-01976-f007:**
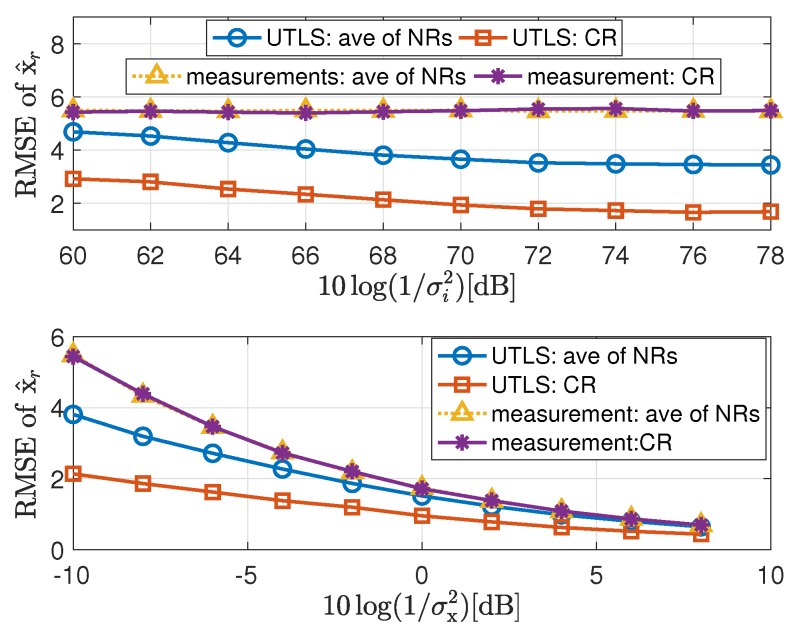
RMSEs of the measurements and the estimations of the locations of the receivers via the UTLS algorithm. In the upper subplot, we assume $10\log(1/\sigma_{x}^{2})=-10$ dB and $10\log\left(1/\sigma_{\mu_{h}}^{2}\right)=10\log\left(1/\sigma_{\mu_{i}}^{2}\right)=96$ dB. In addition, we assume $10\log\left(1/\sigma_{i}^{2}\right)=10\log\left(1/\sigma_{h}^{2}\right)=68$ dB in the lower subplot.

**Figure 8 sensors-19-01976-f008:**
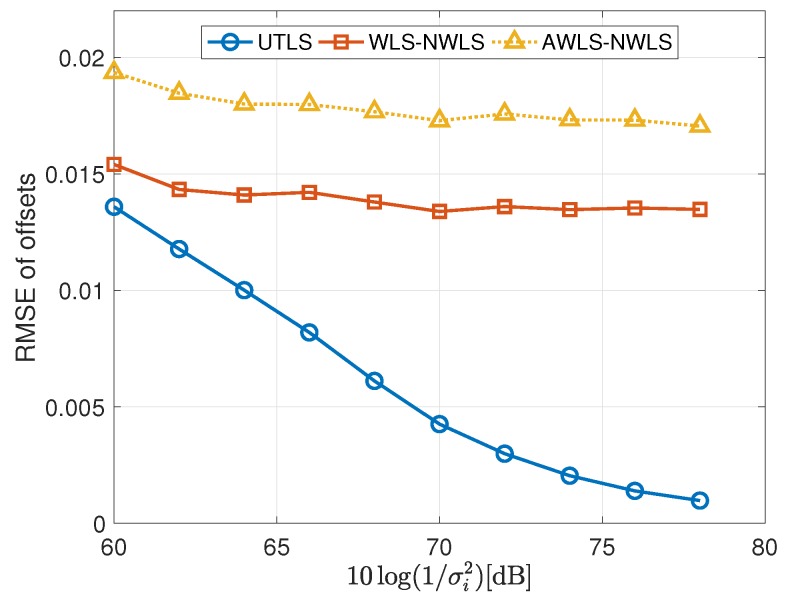
RMSEs of the estimation of the offsets of the receivers via the UTLS, the WLS-NWLS and the AWLS-NWLS algorithms, respectively, with $10\log\left(1/\sigma_{\mu_{h}}^{2}\right)=10\log\left(1/\sigma_{\mu_{i}}^{2}\right)=96$ dB and $10\log(1/\sigma_{x}^{2})=-10$ dB.

**Figure 9 sensors-19-01976-f009:**
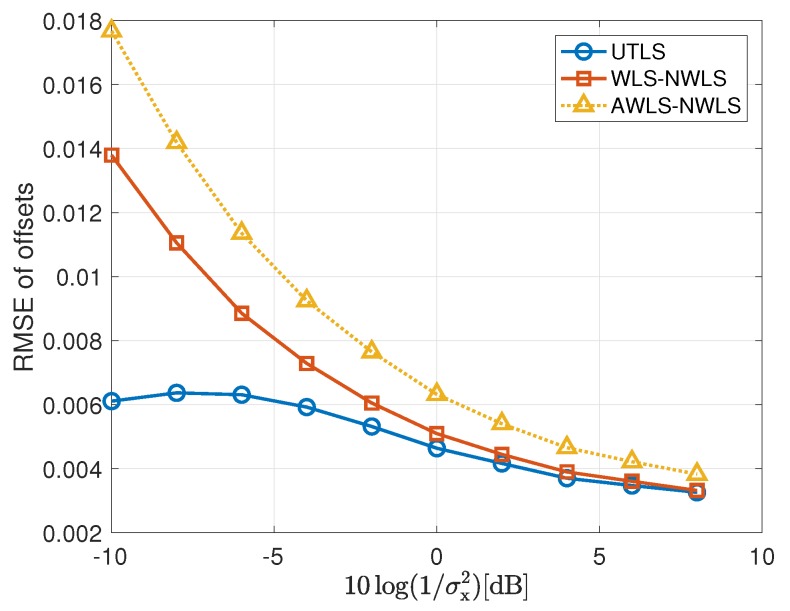
RMSEs of the estimation of the offsets of the receivers via the UTLS, the WLS-NWLS and the AWLS-NWLS algorithms, respectively, with $10\log\left(1/\sigma_{\mu_{h}}^{2}\right)=10\log\left(1/\sigma_{\mu_{i}}^{2}\right)=96$ dB, and $10\log\left(1/\sigma_{i}^{2}\right)=10\log\left(1/\sigma_{h}^{2}\right)=68$ dB.

**Figure 10 sensors-19-01976-f010:**
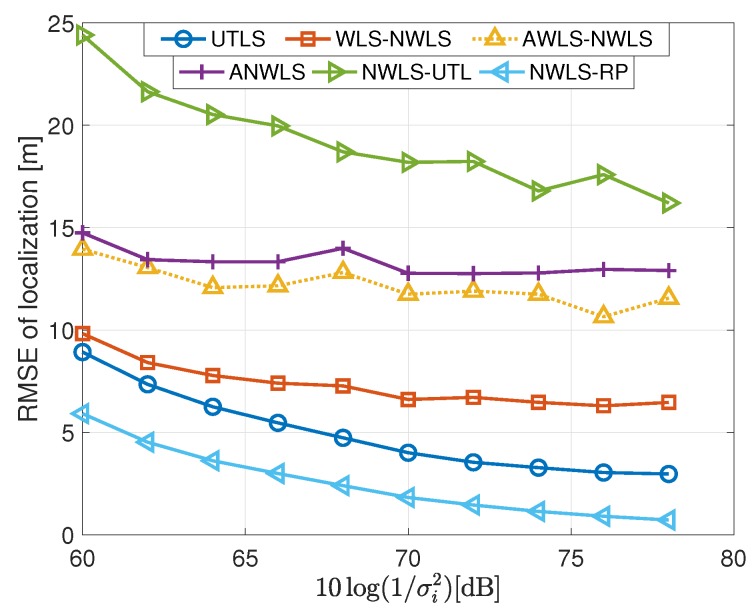
RMSEs of localization via the UTLS, the WLS-NWLS, the AWLS-NWLS, the ANWLS, the NWLS-UTL and the NWLS-RP algorithms, respectively, with $10\log\left(1/\sigma_{\mu_{h}}^{2}\right)=10\log\left(1/\sigma_{\mu_{i}}^{2}\right)=96$ dB and $10\log(1/\sigma_{x}^{2})=-10$ dB.

**Figure 11 sensors-19-01976-f011:**
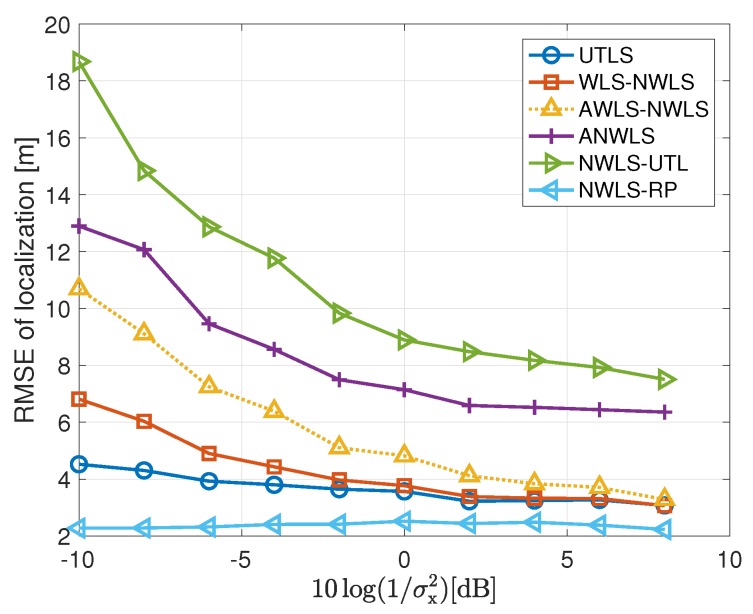
RMSEs of localization via the UTLS, the WLS-NWLS, the AWLS-NWLS, the ANWLS, the NWLS-UTL and the NWLS-RP algorithms, respectively, with $10\log\left(1/\sigma_{\mu_{h}}^{2}\right)=10\log\left(1/\sigma_{\mu_{i}}^{2}\right)=80$ dB and $10\log\left(1/\sigma_{i}^{2}\right)=10\log\left(1/\sigma_{h}^{2}\right)=68$ dB.

**Table 1 sensors-19-01976-t001:** The algorithms in the simulations.

Algorithms	Synchronization		Localization
	Clock Skew	Clock Offset	
UTLS (proposed)	EM	WLS	NWLS
NWLS-UTL [4]	Tailored WLS	*∕*	Tailored NWLS
WLS-NWLS	WLS	WLS	NWLS
AWLS-NWLS	AWLS	WLS	NWLS
ANWLS	1	0	NWLS
NWLS-RP	real value	real value	NWLS

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
