# Peer review of "Underwater Target Localization and Synchronization for a Distributed SIMO Sonar with an Isogradient SSP and Uncertainties in Receiver Locations"

_sensors, 2019, doi:10.3390/s19091976_

Round 1

Reviewer 1 Report

First of all, I want to congratulate the authors for their work presented in this paper. The paper is well-written and easy to follow. They present a very good paper aimed to solve a problem in underwater wireless sensor networks. There are some proposed changes aimed to enhance the quality of the paper:

1 The following papers are based on location and should be included in the introduction to show how the problem of location in other challenging environments have been previously solved:

Seamless outdoors-indoors localization solutions on smartphones: implementation and challenges. ACM Computing Surveys (CSUR) 48 (4), 53, 2016

The Development of Two Systems for Indoor Wireless Sensors Self-location. Ad Hoc & Sensor Wireless Networks 8 (3-4), 235-258, 2009

2 A paper which is focused on the use of acoustic signals for location and clock synchronization should mention the different types of acoustic modems that are available or at least cite a paper focused on this topic. I suggest the following reference:

Underwater acoustic modems. IEEE Sensors Journal, 2016, vol. 16, no 11, p. 4063-4071.

3 The conclusion section is too long. I suggest to move some of the content to the results section and just summarize the important part in the conclusions. In addition, the conclusion section must include future work.

4 Their results point out that NWLS-RP offers better results than their proposed algorithm. As a reviewer, I was expecting to read why it is better their proposed algorithm than NWLS-RP or when it is better to use their algorithm. There are any significant differences between using the proposed algorithm and NWLS-RP in terms of energy use or other important factors? Why we should use the proposed algorithm if the current NWLS-RP offers high her accuracy? In which cases will be better to use the presented algorithm.

5 When the readers see in the results section “In order to fit the statistical properties of the noise samples, 2000 Monte Carlo trails are utilized to evaluate the performance of our proposed algorithms” they are expecting to see in the graphics not only the mean value of the performance but also the standard deviation of values to really know if there are differences between the data or not. I suggest showing this type of data, including the standard deviation of another statistical factor to show if the results of different algorithms are different or not.

Author Response

The response to Reviewer 1 is in the attachment.  Attachment enclosed please.

Reviewer 2 Report

The submitted manuscript is well-written and interesting. However, in my opinion it cannot be published in this form. The following points should be improved:

The authors should compare their results with other researchers’ work. A lot of research has been conducted in this field, therefore comparing obtained results with benchmark algorithms is insufficient.

The experimental studies are needed. The authors largely based on their previous simulation tests, which have not been verified under real conditions. What is more, a lot of approximation were introduced in the methodology, for example sound speed parameters, ocean current model, initial position of receivers and noise parameters. Due to this fact, the obtained algorithm may be inaccurate.

The authors assume, that a and b coefficient in the equation 1 are constant and known a priori. How are they calculated and are they constant regardless of depth? In my opinion, Mackenzie equation could be used to calculate a sound speed at any depth considering salinity and temperature as well.

The established chain length and distance between receivers were not presented. Additionally, the target depth varies from 200 m to 3000 m, which put the target on a plane with the receivers. Will algorithm work properly under this condition?  

Author Response

The response to Reviewer 2 is in the attachment. Attachment enclosed please.

Round 2

Reviewer 2 Report

I would like to thank the authors for a comprehensive response. I appreciate the work which was undertaken to conduct the research, however, I think that the paper would be much better if the experimental studies were involved. I understand that underwater experiments are very difficult to conduct, but, on the other hand, the authors should verify their plenty of simulation experiments. Since, I believe, I will have a great pleasure to read authors’ papers involving experimental results in the near future.

This manuscript is a resubmission of an earlier submission. The following is a list of the peer review reports and author responses from that submission.

Round 1

Reviewer 1 Report

Paper should be rewritten, removing those parts which are in a previous paper in IEEE Access and present the incremental analysis where uncertainties are considered. If you have performance of the previous method, which is the improvement now? Which is the increase in computational cost?

Reviewer 2 Report

This paper presents an underwater localization system based on a SIMO (Simple Input Multiple Output) approach. It is an evolution of a previous work already published by the authors, that is referenced as [4]. That paper already dealt with the depth-dependent sound speed in a similar environment. The main innovations of this paper are the addressing of uncertainties in receiver locations and the time synchronization into their previous approach. Receiver locations are related to the background flow velocity (BFV), and local time of the receivers are modelled in function of two parameters: clock skews and clock offsets.

The UTLS algorithm consist of an EM (expectation – maximization) algorithm that iteratively approaches the values of time skews of the receivers, as well as their location. Clock offsets are then estimated by Weighted Least Squares (WLS) regression. With these parameters – time and position parameters of the receivers – and the observation vector, location is calculated by means of Nonlinear Weighted Least Squares.

The simulation campaign consisted on 2000 Monte Carlo trails.  The results show a better performance of proposed algorithm in both time parameters and location estimation in front of some common statistical methods

The paper seems to be correct, well written and provides a step forward about the location problem in underwater environments. The novelties introduced in comparison with their previous paper [4] may be enough to justify a new publication.

In Section 4 (Numerical simulations), the UTLS proposed algorithm should be compared with other proposal, such as the ones cited in Section 1. Introduction, or a justification of why these proposals are not comparable should be provided.

The authors must also specify the simulation tool used for these simulations, and justify why 2000 simulations are required. The number of repetitions of experiments is not randomly choosen, it must be related with the variance of the results. The authors should read Montgomery, D.C. (1991) Design and Analysis of Experiments. 3rd Edition, John Wiley & Sons, Inc., New York.

There are some small typographic mistakes to correct, such as:

Line 24: “can provides”

Line 57: “There are some researchers take”

Line 151: “it dose not cause”

Fake references 48 and 49 must be removed.